# Digital Image Correlation with a Prism Camera and Its Application in Complex Deformation Measurement

**DOI:** 10.3390/s23125531

**Published:** 2023-06-13

**Authors:** Hao Hu, Boxing Qian, Yongqing Zhang, Wenpan Li

**Affiliations:** 1Qingdao Research Institute, School of Marine Science and Technology, Northwestern Polytechnical University, Xi’an 710072, China; 2School of Mechanical and Precision Instrument Engineering, Xi’an University of Technology, Xi’an 710048, China; qianboxing1023@126.com; 3Department of Mechanical and Automation Engineering, The Chinese University of Hong Kong, Hong Kong 999077, China

**Keywords:** three-channel color image correlation, sub-pixel matching, deformation measurement, simulation, cylinder compression

## Abstract

Given the low accuracy of the traditional digital image correlation (DIC) method in complex deformation measurement, a color DIC method is proposed using a prism camera. Compared to the Bayer camera, the Prism camera can capture color images with three channels of real information. In this paper, a prism camera is used to collect color images. Relying on the rich information of three channels, the classic gray image matching algorithm is improved based on the color speckle image. Considering the change of light intensity of three channels before and after deformation, the matching algorithm merging subsets on three channels of a color image is deduced, including integer-pixel matching, sub-pixel matching, and initial value estimation of light intensity. The advantage of this method in measuring nonlinear deformation is verified by numerical simulation. Finally, it is applied to the cylinder compression experiment. This method can also be combined with stereo vision to measure complex shapes by projecting color speckle patterns.

## 1. Introduction

As a numerical calculation method based on speckle images, digital image correlation [1] has been widely used to measure the deformation and strain field on the stressed surface. Moreover, in combination with binocular stereo vision, a three-dimensional shape [2] can also be obtained by projecting a speckle pattern on the measured surface. The classic digital image correlation takes gray speckle images as the basic data, and its algorithm mainly includes integer-pixel matching and sub-pixel matching. Sub-pixel matching is the key to obtaining accurate measurement data. Its essence is nonlinear optimization, usually solved iteratively by the Newton method. The mainstream sub-pixel matching algorithms are the forward accumulation Gauss-Newton method (FA-GN) [3] and the reverse combination Gauss-Newton method (IC-GN) [4]. Both have the same accuracy. Compared to the IC-GN algorithm, FA-GN is more flexible in selecting shape function and correlation function.

The digital image correlation method based on gray speckle images aims to improve accuracy and speed. There is much theoretical research on the quality evaluation of speckle images, shape functions, interpolation functions, optimization methods, subset sizes, strain field solutions, computational efficiency, and other technical details. Since the theory is relatively mature [5], the current research direction is mainly the application of underwater, high temperature, high light, high speed, rotation, large field of view, micro-scale, and other complex scenes [6].

Due to the popularity of color cameras, the method of measuring deformation and shape based on color speckle images has received enough attention and has even been applied to high-temperature deformation [7], the topography of a liquid interface [8], and monitoring of painting [9]. In fact, researchers are interested in the rich information contained in the color image. Currently, the measurement methods using color speckle images are divided into three categories. The first is to use a Bayer color camera (also known as Color Filter Array, CFA) to obtain color images. In Bayer color cameras, only one of the color values of red, green, and blue is really from CCD, and the other two values are calculated by interpolation, that is, the estimated value. Therefore, this category mainly studies color image preprocessing and interpolation methods, hoping to improve measurement accuracy [10,11]. The second is to build a complex optical path to obtain more color information through optical principles. This category is mainly carried out in the laboratory and is used for the three-dimensional reconstruction of small object surfaces [12,13,14,15,16,17]. The idea is to separate the red and blue channels of a single-color camera using a light-splitting prism, and then form binocular stereo vision through reflectors [18]. In addition, Luis et al. [19] synthesized the characteristic stripe and speckle pattern into color coding. They improved the measurement accuracy of 3D displacement and in-plane displacement due to the addition of phase information.

The third is to study the matching algorithm of color images, mainly improving the classic DIC algorithm to apply to color images. In 2003, Yonyama et al. [20] used NCC (normalized cross correlation) function as the three-channel correlation function, confirmed that a smaller subset can be used to match, and measured the displacement of rigid body rotation. In 2015, Ghulam et al. [21] used the NSSD (normalized sum of squared differences) function to apply color speckle images to small strain measurements of deformable solids. Subsequently, the different sizes of color speckle particles and different deformation were analyzed in detail, and it was believed that the color DIC had a better effect than the gray DIC [22]. However, the above methods do not specifically discuss the process of the algorithm, nor do they consider the changes in light intensity on different channels before and after deformation. As a result, the algorithm cannot be applied in the situation of complex changes in ambient light. Recently, Wang [23] proposed a correlation function based on hue, which is robust to the scaling and shifting of light intensity. This method can be used to measure the scaling deformation and rotation deformation between large frames, but the implementation is complex. Since the prism camera can provide accurate pixel color values on three channels, it has more advantages than the Bayer color camera in image matching [24]. In this paper, a prism camera is used to collect color speckle images. Based on the classic gray image matching algorithm, the light intensity changes of the three channels before and after deformation are comprehensively considered, and a correlation matching algorithm based on color images is derived. Through numerical simulation and real experiments, its effectiveness in solving complex deformation problems is verified.

## 2. Algorithm

### 2.1. Principle

The traditional gray image matching algorithm, on the one hand, can ensure that the subset has enough information, especially when the contrast of the speckle image is not strong, and usually selects a large reference subset. On the other hand, because the second-order shape function contains many unknown coefficients, which will lead to a large standard deviation of the calculation results, the first-order shape function is often used in the matching algorithm. However, when matching complex deformation, the mapping relationship between the large reference subset and the matching subset presents non-linear, and its deformation law is difficult to be described by the first-order shape function. Even the second-order shape function may not be accurately described. In this case, making match calculations will lead to large matching errors and inaccurate measurement results.

Because of the low accuracy of the traditional correlation matching in complex deformation measurement, based on the abundant information of the three channels of the color image, a correlation matching algorithm using the combination of the brightness information of the three channels of a color image is proposed based on the FA-GN algorithm. It includes integer-pixel matching, sub-pixel matching, and initial estimation of light intensity.

The basic principle is shown in Figure 1. With the help of the brightness information of the three channels of the prism camera, select a smaller subset at the same position of each channel. Thus, the three small subsets can jointly provide enough brightness information to implement matching. At the same time, even for complex deformation, the deformation of each small subset can obey the first-order shape function. In the case of measuring complex deformations, it avoids large iteration residuals in the traditional correlation matching method caused by the following two aspects: it is difficult to provide sufficient matching information when using a small subset; the shape function is difficult to describe the deformation law when using a large subset.

### 2.2. Integer-Pixel Matching

First, the corresponding position of the integer-pixel level on the reference image and the deformed image should be determined. The sum of the grayscale of the reference subset and the deformation subset is abbreviated as follows, and the like:(1)$$\left\{\begin{array}{c} \sum_{s=1}^{3}\sum_{i=-m}^{m}\sum_{j=-m}^{m}f^{s}(x_{i},y_{j})\to \sum_{i=1}^{3\times {\left(2m+1\right)}^{2}}f_{i}^{s}\to \sum f\\ \sum_{s=1}^{3}\sum_{i=-m}^{m}\sum_{j=-m}^{m}g^{s}(x_{i}^{\prime },y_{j}^{\prime })\to \sum_{i=1}^{3\times {\left(2m+1\right)}^{2}}g_{i}^{s}\to \sum g \end{array}\right.$$
where, $\left(x_{i},y_{j}\right)$ and $\left(x_{i}^{\prime },y_{j}^{\prime }\right)$ are the pixel coordinates in the reference subset and the matching subset, respectively. $f^{s}(x_{i},y_{j})$ and $g^{s}(x_{i}^{\prime },y_{j}^{\prime })$ are the grayscales of the corresponding positions on the *s* channel in the color image. *m* is the half-length of the subset in pixels.

The correlation function used for integer-pixel matching is the ZNSSD (zero-normalized sum of squared differences) function:(2)$$C_{ZNSSD}=\sum_{i=1}^{3\times {\left(2m+1\right)}^{2}}{\left[\frac{f-f_{\mu }}{\sqrt{\sum {\left(f-f_{\mu }\right)}^{2}}}-\frac{g-g_{\mu }}{\sqrt{\sum {\left(g-g_{\mu }\right)}^{2}}}\right]}^{2}$$
where, $f_{\mu }$ and $g_{\mu }$ are the gray mean values of the reference subset and the deformation subset on the three channels:(3)$$\left\{\begin{array}{c} f_{\mu }=\frac{1}{3\times {\left(2m+1\right)}^{2}}\sum_{s=1}^{3}\sum_{i=-m}^{m}\sum_{j=-m}^{m}f^{s}(x_{i},y_{j})\\ g_{\mu }=\frac{1}{3\times {\left(2m+1\right)}^{2}}\sum_{s=1}^{3}\sum_{i=-m}^{m}\sum_{j=-m}^{m}g^{s}(x_{i}^{\prime },y_{j}^{\prime }) \end{array}\right.$$

The calculation amount of integer-pixel matching increases exponentially with the increase in subset size. Therefore, a fast Integer-pixel search method is proposed. It is believed that the gray difference of the corresponding position before and after the deformation will not be greater than 50 on the three channels. With this constraint, some impossible positions in the search area can be eliminated. For the remaining effective pixel positions, use the calculation process in Figure 2. First, the similarity is calculated using small areas, and these impossible matching positions whose correlation coefficient is greater than the average of the overall correlation coefficient ${\bar{C}}_{m}$ are excluded. Then, increase the size of the subset, calculate the correlation coefficient and its average value for the remaining area, and remove some impossible pixels again. This is repeated until the last pixel position is left. Usually, when half of the length of the subset is less than five pixels, the search is completed.

### 2.3. Sub-Pixels Matching

After the integer-pixel matching is completed, the corresponding positions of the sub-pixel level of the reference image and the deformed image should be determined. It is considered that the gray level of any point in the subset changes according to the linear model before and after deformation. The minimum distance square sum function including the linear light intensity coefficient is selected to construct goal expression:(4)$$F(p)=\sum_{s=1}^{3}\sum_{i=-m}^{m}\sum_{j=-m}^{m}{\left[a_{s}\times f^{s}(x_{i},y_{j})+b_{s}-g^{s}(x_{i}^{\prime },y_{j}^{\prime })\right]}^{2}$$
where, $a_{s}$ and $b_{s}$ are the coefficients of light intensity in the subset of the *s*-th channel. Subset’s deformation uses first-order shape function:(5)$$\left\{\begin{array}{c} x_{i}^{\prime }=x_{i}+u+u_{x}\Delta x+u_{y}\Delta y\\ y_{j}^{\prime }=y_{j}+v+v_{x}\Delta x+v_{y}\Delta y \end{array}\right.$$
where, $\left(\Delta x,\Delta y\right)$ is the offset of the calculated point in the subset in the horizontal and vertical directions relative to the subset center. Then, the unknown vector including light intensity coefficient $\left(a_{1},b_{1},a_{2},b_{2},a_{3},b_{3}\right)$ and deformation coefficient $\left(u,u_{x},u_{y},v,v_{x},v_{y}\right)$ is:(6)$$p={\left[u,u_{x},u_{y},v,v_{x},v_{y},a_{1},b_{1},a_{2},b_{2},a_{3},b_{3}\right]}^{T}$$

The calculation of the deformation coefficient in the correlation function is the unconstrained extremum problem of the multivariate function. For the multivariate function F(p), the iterative format is obtained by the Newton method:(7)$$p^{\left(n+1\right)}=p^{\left(n\right)}-{\left(\nabla^{2}F(p^{\left(n\right)})\right)}^{-1}\nabla F(p^{\left(n\right)})$$
where, ∇F(p) and $\nabla^{2}F(p)$ are first-order and second-order partial derivative matrix, respectively:(8)$$\nabla F(p)={\left[\begin{array}{cccc} \frac{\partial F}{\partial p_{1}} & \frac{\partial F}{\partial p_{2}} & \cdots & \frac{\partial F}{\partial p_{12}} \end{array}\right]}^{T}$$
(9)$$\nabla^{2}F(p)=\left[\begin{array}{cccc} \frac{\partial^{2}F}{\partial p_{1}^{2}} & \frac{\partial^{2}F}{\partial p_{1}\partial p_{2}} & \cdots & \frac{\partial^{2}F}{\partial p_{1}\partial p_{12}}\\ \frac{\partial^{2}F}{\partial p_{2}\partial p_{1}} & \frac{\partial^{2}F}{\partial p_{2}^{2}} & \cdots & \frac{\partial^{2}F}{\partial p_{2}\partial p_{12}}\\ \vdots & \vdots & \frac{\partial^{2}F(p)}{\partial p_{k}\partial p_{l}} & \vdots \\ \frac{\partial^{2}F}{\partial p_{12}\partial p_{1}} & \frac{\partial^{2}F}{\partial p_{12}\partial p_{2}} & \cdots & \frac{\partial^{2}F}{\partial p_{12}^{2}} \end{array}\right]$$

In the partial derivative matrix, for six deformation coefficients, when k,l = 1, 2, …, 6:(10)$$\left\{\begin{array}{l} \frac{\partial F(p)}{\partial p_{k}}=2\sum_{s=1}^{3}\sum_{i=-m}^{m}\sum_{j=-m}^{m}\left[a_{s}\times f^{s}(x_{i},y_{j})+b_{s}-g^{s}(x_{i}^{\prime },y_{j}^{\prime })\right]\frac{\partial g^{s}}{\partial p_{k}}\\ \frac{\partial^{2}F(p)}{\partial p_{k}\partial p_{l}}\approx 2\sum_{s=1}^{3}\sum_{i=-m}^{m}\sum_{j=-m}^{m}\frac{\partial g^{s}}{\partial p_{k}}\frac{\partial g^{s}}{\partial p_{l}} \end{array}\right.$$

Abbreviation $\frac{\partial g(x_{i}^{\prime },y_{j}^{\prime })}{\partial x^{\prime }}=g_{x^{\prime }}$,$\frac{\partial g(x_{i}^{\prime },y_{j}^{\prime })}{\partial y^{\prime }}=g_{y^{\prime }}$, then
(11)$$\left\{\begin{array}{c} \frac{\partial g^{s}}{\partial p_{1}}=g_{x^{\prime }}^{s},\frac{\partial g^{s}}{\partial p_{2}}=g_{x^{\prime }}^{s}\Delta x,\frac{\partial g^{s}}{\partial p_{3}}=g_{x^{\prime }}^{s}\Delta y\\ \frac{\partial g^{s}}{\partial p_{4}}=g_{y^{\prime }}^{s},\frac{\partial g^{s}}{\partial p_{5}}=g_{y^{\prime }}^{s}\Delta x,\frac{\partial g^{s}}{\partial p_{6}}=g_{y^{\prime }}^{s}\Delta y \end{array}\right.$$

In the partial derivative matrix, for six light intensity coefficients, that is, when *k* = 7, …, 12:(12)$$\left\{\begin{array}{l} \frac{\partial F(p)}{\partial a_{s}}=2\sum_{i=-m}^{m}\sum_{j=-m}^{m}\left[a_{s}\times f^{s}(x_{i},y_{j})+b_{s}-g^{s}(x_{i}^{\prime },y_{j}^{\prime })\right]f^{s}(x_{i},y_{j})\\ \frac{\partial F(p)}{\partial b_{s}}=2\sum_{i=-m}^{m}\sum_{j=-m}^{m}\left[a_{s}\times f^{s}(x_{i},y_{j})+b_{s}-g^{s}(x_{i}^{\prime },y_{j}^{\prime })\right] \end{array}\right.$$

The calculation flow of sub-pixel matching based on Newton’s iteration is shown in Figure 3. Newton iteration needs to determine the exact initial value of the integer-pixel position and unknown coefficient in advance and set the termination condition. It can be set that in the adjacent iteration, the iteration is terminated when the modulus of the unknown coefficients change is less than 0.001.

The initial value of iteration should be as accurate as possible because Newton’s iteration method is easy to fall into the local optimal solution. The initial value of the deformation coefficient is generally zero. In the actual measurement, due to the interference of ambient light, the brightness of the image on the three channels before and after deformation may change. To ensure the stability of sub-pixel matching, the initial values of light intensity changes before and after deformation in each channel are solved before iteration. Take one of the channels as an example to illustrate the solution.

### 2.4. Initial Value Estimation of Light Intensity Coefficients

After the completion of integer-pixel matching, *n* × *n* pixel regions centered on the reference node $f_{5}$ and matching node $g_{5}$ are selected on the reference image and deformed image, as shown in Figure 4. $f_{i}$ and $g_{i}$ are the gray levels of corresponding positions. Then, according to the linear transformation on light intensity, it can be obtained:(13)$$\left[\begin{array}{cc} f_{1} & 1\\ f_{2} & 1\\ \vdots & \vdots \\ f_{i} & 1 \end{array}\right]\left[\begin{array}{c} a\\ b \end{array}\right]=\left[\begin{array}{c} g_{1}\\ g_{2}\\ \vdots \\ g_{i} \end{array}\right]\;(\mathrm{Abbreviated}\;\mathrm{as}\;f{\left[a,b\right]}^{T}=g)$$

Its least squares solution is:(14)$${\left[a,b\right]}^{T}={\left(f^{T}f\right)}^{-1}f^{T}g$$

The initial estimation of light intensity is generally selected as a 5 × 5-pixel region.

If the light intensity changes of the three channels before and after deformation are not considered, the sub-pixel matching algorithm in Section 2.3 can also be derived based on IC-GN algorithm. The specific derivation process is shown in Appendix A. In some measurement scenes where ambient lighting is uniform and stable, the sub-pixel matching algorithm in Appendix A has higher computational efficiency.

## 3. Numerical Simulation

### 3.1. Generate Speckle Image

The speckle images before and after deformation are generated by computer simulation to verify the effectiveness of this method [25]. The gray image I(x,y) and $I^{\prime }(x,y)$ of each channel of color speckle image before and after deformation are generated according to the following formula:(15)$$\left\{\begin{array}{c} I(x,y)=\sum_{k=1}^{r}A_{k}\exp\left(-\frac{{\left(x-x_{k}\right)}^{2}+{\left(y-y_{k}\right)}^{2}}{{\left(D/2\right)}^{2}}\right)\\ I^{\prime }(x,y)=\sum_{k=1}^{r}A_{k}\exp\left(-\frac{{\left(x-x_{k}^{\prime }\right)}^{2}+{\left(y-y_{k}^{\prime }\right)}^{2}}{{\left(D/2\right)}^{2}}\right) \end{array}\right.$$
where, *D* is the diameter of speckle particles, and *r* is the total number of speckle particles.$\left(x_{k},y_{k}\right)$ and *A_k_* is the position and strength of the *k*-th speckled particle. $\left(x_{k}^{\prime },y_{k}^{\prime }\right)$ is the position of the *k*-th speckle particle after translation. It is described by displacement function before and after translation:(16)$$\left\{\begin{array}{c} x_{k}^{\prime }=x_{k}+u(x,y)\\ y_{k}^{\prime }=y_{k}+v(x,y) \end{array}\right.$$
where, u(x,y) and v(x,y) are displacement functions in horizontal and vertical directions, respectively, which should be continuous. To prove the superiority of this method in dealing with complex deformation, the displacement functions u(x,y) and v(x,y) are defined as sine laws [26] so that the resulting deformation is nonlinear and non-uniform. *T* is the period and α is the amplitude:(17)$$\left\{\begin{array}{c} u(x,y)=\alpha \sin(2\pi x/T)\sin(2\pi y/T)\\ v(x,y)=\alpha \cos(2\pi x/T)\cos(2\pi y/T) \end{array}\right.$$

In the algorithm test in this paper, the image size is 400 × 400 pixels^2^, the particle size is 4 pixels, and the particle number is 7000. On the reference image of each channel, a displacement function with a period *T* of 20 and amplitude α of 0.5 pixels is applied first. Then, add the light intensity transformation with (a, b) of (0.95, 8), (0.93, 12), and (0.94, 10) on the three channels, respectively. Finally, color speckle images before and after deformation are synthesized. The color reference image is shown in Figure 5a. To compare with the traditional method, the gray speckle image is obtained by graying the color image before and after deformation. The weighted average factor used for graying is (0.299, 0.578, 0.114). The grayscale reference image is shown in Figure 5b. An improved algorithm (called color FA-GN) is used on color speckle images, and a traditional algorithm (called gray FA-GN) is used on gray speckle images. Both choose bilinear interpolation and first-order shape function. To ensure that the amount of information involved in matching the subset is basically the same in the two algorithms, seven groups of different subset sizes in Table 1 are selected for comparative calculation. Using the Intel Core i7 processor, run the computer with 16G memory to execute the written program.

In Figure 5b, the calculation area (red box) and the distribution of calculation nodes (blue cross) of the two methods are also marked. The horizontal and vertical distance (step length) between adjacent nodes is 15 pixels, the horizontal coordinate is 41:15:341, and the vertical coordinate is 49:15:349. The scale of the nodes is 21 × 21. Each node is calculated independently. The size of the subset shown in the green box is 23 × 23 pixels^2^.

### 3.2. Result Analysis

In the DIC algorithm, the size of the subset is closely related to the calculation deviation. In general, the larger the subset, the more information it contains, and the more accurate the corresponding matching result [27]. To ensure fairness, in Table 1, the number of pixels contained in the subset used by the two methods in each group is approximately the same. The calculation results of these two methods are compared from not only the calculation deviation of the U-field and V-field but also the average number and computing time of iterations. The calculated deviation is calculated from two aspects: mean absolute error (MAE) and root mean square error (RMSE):(18)$$\left\{\begin{array}{l} MAE=\frac{1}{n}\overset{n}{\underset{i=1}{\sum }}\left|deviation_{i}\right|=\frac{1}{n}\overset{n}{\underset{i=1}{\sum }}\left|calculated_{i}-ideal_{i}\right|\\ RMSE=\sqrt{\frac{1}{n}\overset{n}{\underset{i=1}{\sum }}{\left(deviation_{i}-deviation_{\mu }\right)}^{2}} \end{array}\right.$$
where, $calculated_{i}$, $ideal_{i}$, and $deviation_{i}$ are the calculated value, ideal value, and deviation of the displacement of the *i*-th node, respectively, and $deviation_{\mu }$ is the average value of the deviation of all nodes.

Draw the deviations in Table 1 into a curve, as shown in Figure 6. In group (1), the calculation deviation of color DIC is larger than that of gray DIC. When the subset is small, the deformation in the subset is close to a linear deformation. Theoretically, the calculation results should be consistent when dealing with linear deformation. However, due to the more unknowns solved by color DIC compared with gray DIC, resulting in greater displacement deviation. However, this deviation is generally within the acceptable range. Moreover, to ensure the stability of the iteration, the selected subset normally will not be too small. With the increase of the size of the gray subset, the deformation in the subset gradually presents nonlinear. Currently, in gray DIC, the first-order shape function is increasingly difficult to describe the deformation law in the subset, resulting in increasing calculation deviation continuously [28]. However, the size of the color subset does not change much, so the calculation deviation remains low. In terms of time and iteration times, the two are approximate. The color DIC algorithm is slightly higher because it spends more time on frequent image access in three channels. In practical applications, the initial values of the deformation and light intensity coefficients can be transferred to surrounding nodes through the seed node, which can greatly shorten the calculation time [29].

Figure 7a,c show the original displacement field applied on the reference image in the numerical simulation, and Figure 7b,d show the horizontal and vertical displacement fields calculated by group (4) using color FA-GN. Compared with the distribution of the ideal displacement field on the left, it can reflect the correctness of the matching.

Still using data from group (4), compare the displacement deviation of the U-field and V-field under the two calculation methods, and the results are shown in Figure 8. The reference subset selected for the color image is small, and the deformation of each subset is approximately linear, so the displacement deviation at each node is small and consistent. However, due to the large subset selected by the traditional method, the deformation in the subset conflicts with the first-order shape function, resulting in a large matching error. Especially when the node is located at the peak and trough of the deformation function, the conflict is the most severe, so the displacement error at these positions is the largest.

## 4. Application

The compression characteristics of high elastic polymer materials, such as rubber, are the focus of elastic mechanics and the material industry. In this paper, a simple experimental device is built, and the experimental schematic is shown in Figure 9. The speckle images of the annular specimen under different static loads are taken by the color camera, and the surface deformation of the specimen under each state is measured. The loading device is a small vise, which manually rotates the screw to apply pressure. The deformation of the specimen after compression is complex, which can verify the feasibility of the color digital image correlation method for measuring large nonlinear deformation.

The experimental site is shown in Figure 10. The outer diameter of the annular rubber is 70 mm, the inner diameter is 12 mm, and the thickness is 20 mm. Before the experiment, a thin layer of matte white paint was covered on the rubber ring surface, and then three colors of red, green, and blue were randomly sprayed, respectively. The model of the prism camera is AT-200GE. It uses three ICX274AL CCD imaging sensors, and the resolution of each channel is 1620 × 1236 pixels, with a pixel size of 4.4 μm. Select a fixed-focus lens with a focal length of 12 mm and object distance is 180 mm. By converting the diameter of the marker circle on the reference image, the pixel equivalent is 3.64 × 10^−2^ mm/pixel.

The deformation of the rubber surface under different stress conditions is measured using the proposed matching method. The subset size is 41 × 41 pixels^2^ and the step length is 10 pixels. Due to space limitations, four compression states are chosen in Figure 11a. The calculated deformation field in horizontal and vertical directions are shown in Figure 11b–e, respectively. The maximum and minimum displacements in the horizontal and vertical directions in each state are shown in the lower right corner of each figure. It can be clearly seen that the measurement results are smooth, and the distribution trend is correct. This reflects the value of the color image-matching algorithm in practical applications.

## 5. Conclusions

Relying on the rich information content of three channels of color images with a prism camera, an improved digital image correlation method to measure complex deformation is proposed in this paper. Considering the interference of the linear change of brightness of three channels before and after deformation in practical applications, a specific matching algorithm is given. It uses the information of three channels at the same location to reduce the reference subset. Therefore, using a second-order shape function with many parameters is avoided, and the matching accuracy and stability are improved. Through numerical simulation, the advantages of this method in measuring complex deformation are confirmed. Combined with binocular stereo vision, this method can also measure complex contour profiles by projecting color speckle patterns. In addition, it also provides a reference for measuring complex contour profiles by projecting multiple gray speckle images.

## Figures and Tables

**Figure 1 sensors-23-05531-f001:**
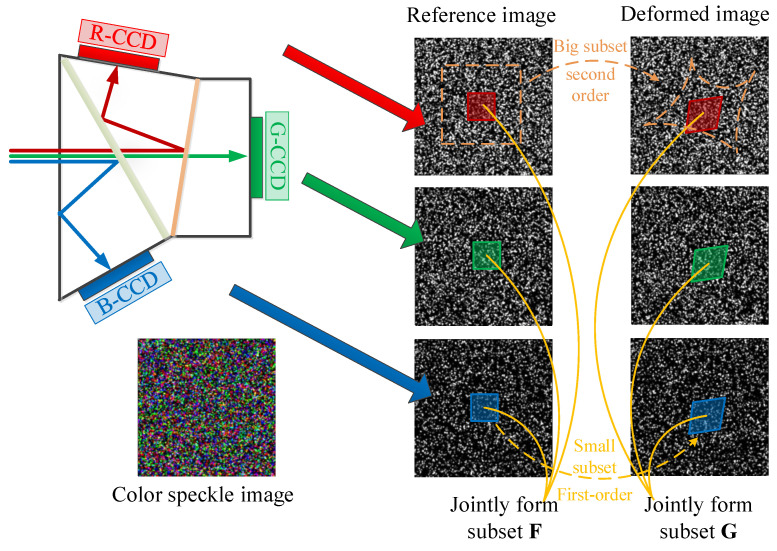
Matching principle of color speckle image.

**Figure 2 sensors-23-05531-f002:**
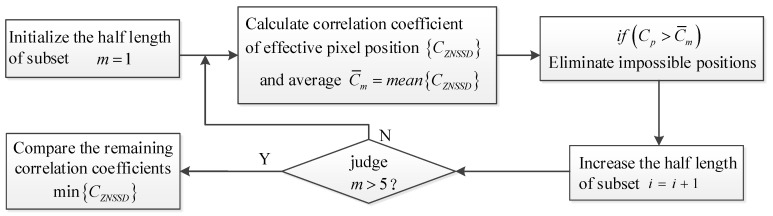
Integer-pixel search process.

**Figure 3 sensors-23-05531-f003:**
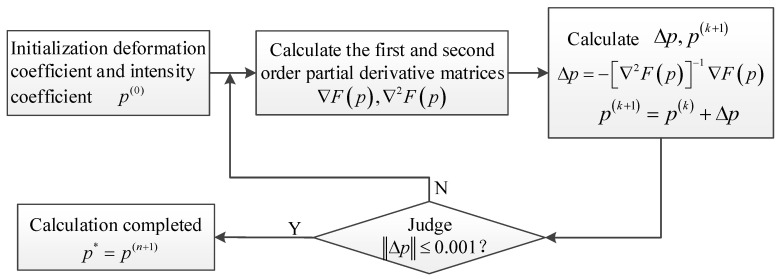
Sub-pixel search calculation process.

**Figure 4 sensors-23-05531-f004:**
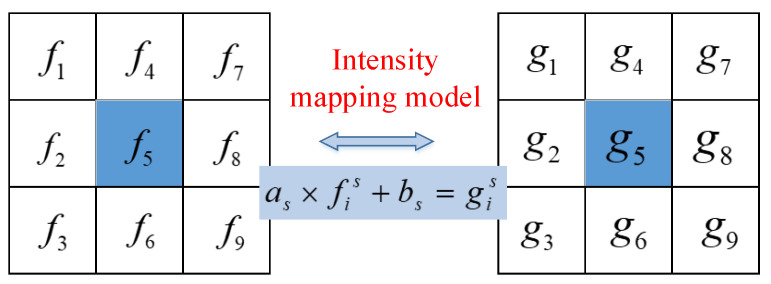
Pixel region of reference image corresponding to deformed image.

**Figure 5 sensors-23-05531-f005:**
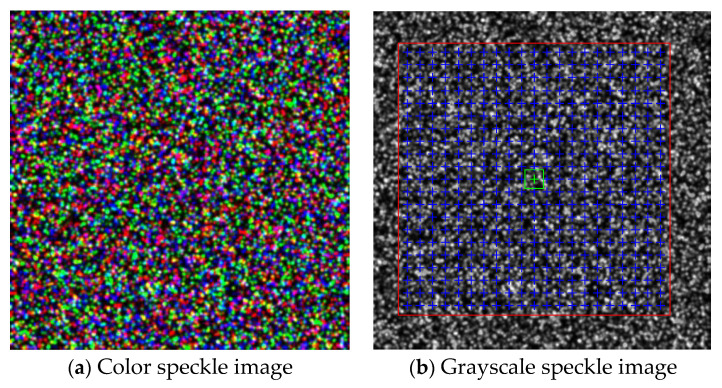
Reference image generated by simulation.

**Figure 6 sensors-23-05531-f006:**
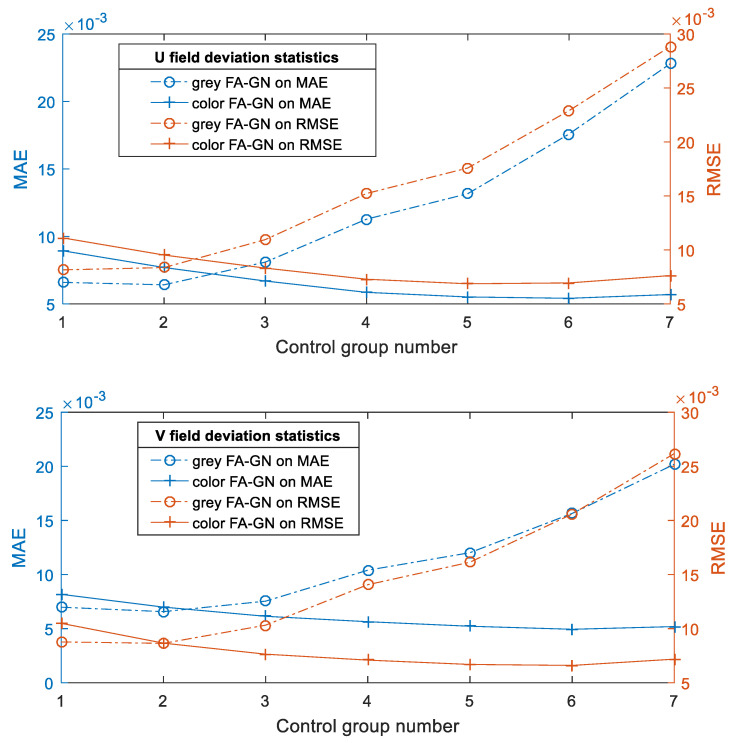
Deviation statistics by biaxial diagram.

**Figure 7 sensors-23-05531-f007:**
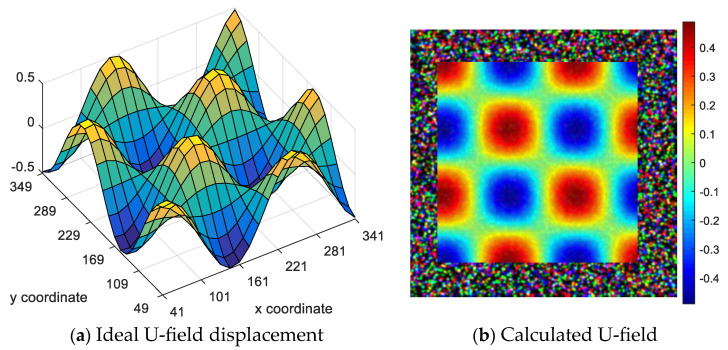
Deformation display.

**Figure 8 sensors-23-05531-f008:**
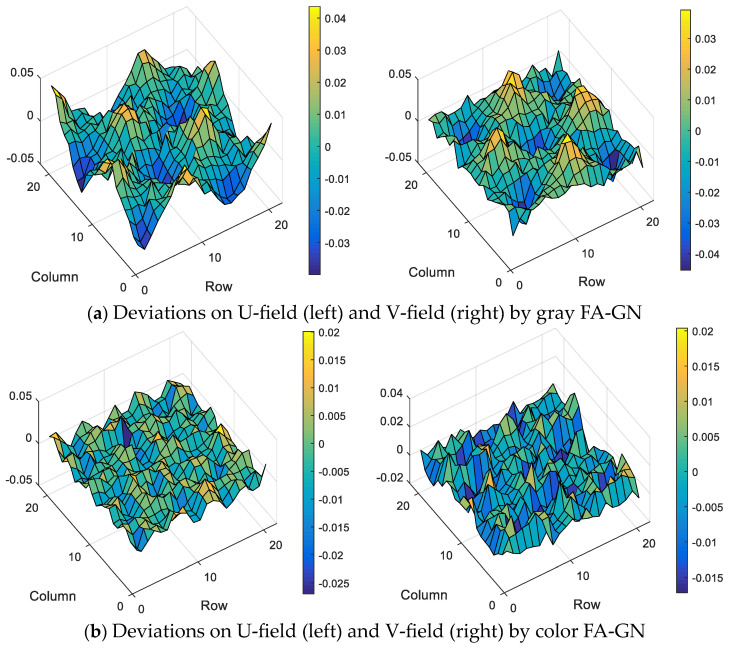
Displacement deviation of nodes by two methods.

**Figure 9 sensors-23-05531-f009:**
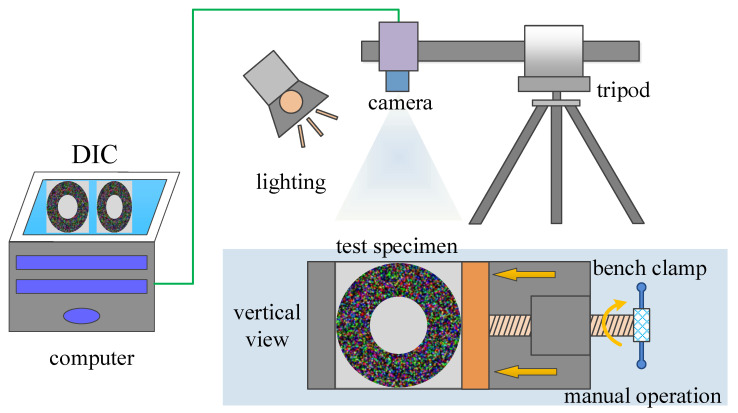
Experiment of cylinder radial compression.

**Figure 10 sensors-23-05531-f010:**
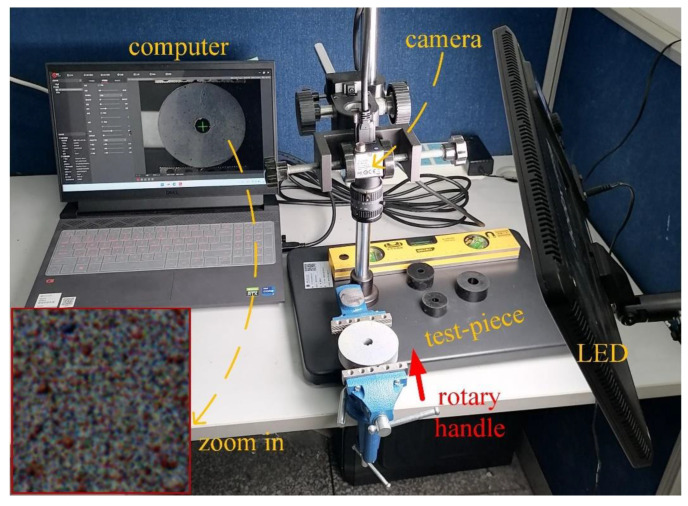
Experimental scene.

**Figure 11 sensors-23-05531-f011:**
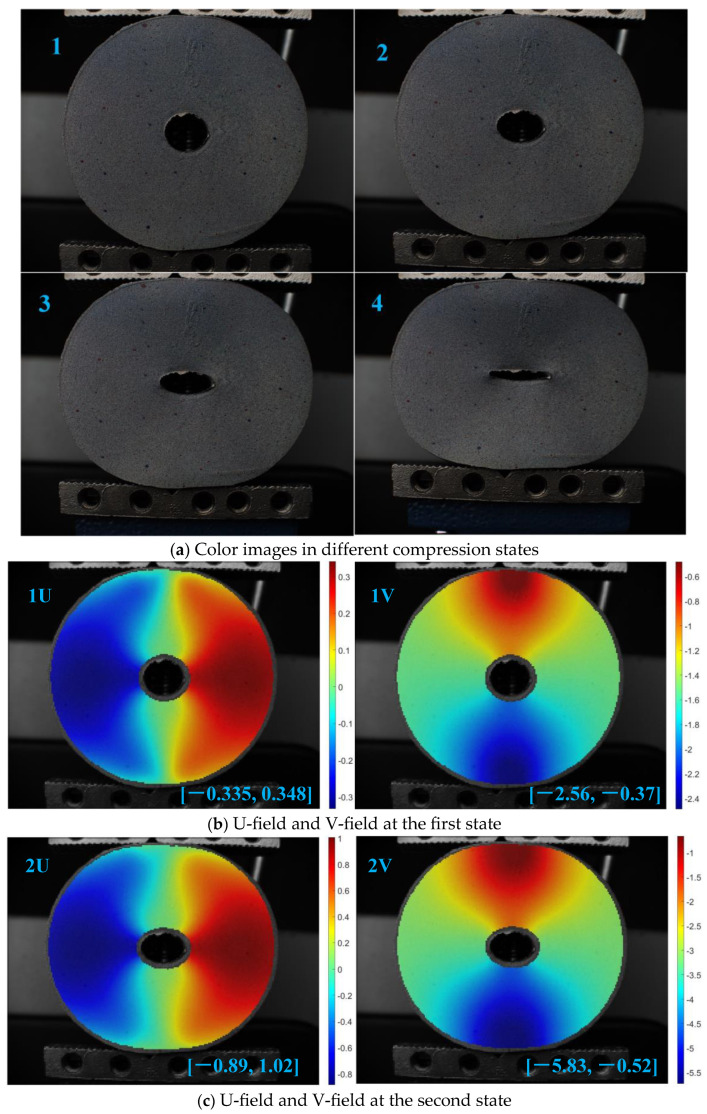
Measurement results of cylinder compression.

**Table 1 sensors-23-05531-t001:** Results of the two methods under different subset sizes.

Group Number	Subset Size	U-Field Displacement Deviation Statistics (×10^−3^)		V-Field Displacement Deviation Statistics (×10^−3^)		Average Value of Iterations	
		MAE	RMSE	MAE	RMSE	Time (×10^−4^ s)	Times
(1)	(2 × 9 + 1)^2^	6.60	8.16	6.98	8.77	5.60	4.92
	(2 × 5 + 1)^2^ × 3	8.94	11.12	8.17	10.49	9.25	5.66
(2)	(2 × 11 + 1)^2^	6.42	8.37	6.57	8.64	6.51	4.78
	(2 × 6 + 1)^2^ × 3	7.70	9.53	6.99	8.66	9.57	5.39
(3)	(2 × 13 + 1)^2^	8.10	10.96	7.54	10.32	7.81	4.68
	(2 × 7 + 1)^2^ × 3	6.70	8.32	6.14	7.64	10.85	5.25
(4)	(2 × 15 + 1)^2^	11.28	15.23	10.40	14.06	9.36	4.68
	(2 × 8 + 1)^2^ × 3	5.86	7.29	5.64	7.09	13.83	5.12
(5)	(2 × 16 + 1)^2^	13.17	17.59	12.02	16.14	10.21	4.70
	(2 × 9 + 1)^2^ × 3	5.52	6.87	5.22	6.69	15.17	5.02
(6)	(2 × 18 + 1)^2^	17.59	22.90	15.65	20.59	11.58	4.64
	(2 × 10 + 1)^2^ × 3	5.42	6.95	4.94	6.60	17.06	4.98
(7)	(2 × 20 + 1)^2^	22.80	28.84	20.19	26.18	13.90	4.62
	(2 × 11 + 1)^2^ × 3	5.70	7.63	5.19	7.18	18.57	4.94

## Data Availability

Not applicable.

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
