# Peer review of "Digital Image Correlation with a Prism Camera and Its Application in Complex Deformation Measurement"

_sensors, 2023, doi:10.3390/s23125531_

Round 1
Reviewer 1 Report
The paper discusses color digital image correlation for in-plane deformation measurement using a prism color camera rather than the usual Bayer-filter one. Detailed simulations and an experiment is presented to convincingly prove the effectiveness of the method. The research is nicely executed and the paper is expected to be of great practical interest.
If I understand correctly, the main novelty of the paper is the use of the prism camera. It would be nice to stress it more in the title and the abstract.
My question is that how the sample color or reflection inhomogeneities affect the measurements.
The paper is well organised, written in good English, however, a spell checking and minor improvements of style is recommended. Moreover, define abbreviations NCC and NSSD (p. 2) and ZNSSD (p. 3). Ref. 11 has no journal indicated. Writing the exponents in "in-line" form (such as 1e-3) is not appropriate.
.
Author Response
(1) If I understand correctly, the main novelty of the paper is the use of the prism camera. It would be nice to stress it more in the title and the abstract.
Reply: Yes, the novelty of this paper lies in the use of a prism camera that can capture speckle images from three channels at once. This feature is utilized and corresponding matching algorithms are proposed. As suggested, both title and abstract are modified.
(2) My question is that how the sample color or reflection inhomogeneities affect the measurements.
Reply: The research in this article mainly focuses on the deformation measurement on the surface of small objects in a laboratory environment. Therefore, there will not be severe reflection inhomogeneities. In addition, to ensure the reliability of the measurement, two light intensity coefficients are added to matching algorithm, which can resist light intensity changes before and after deformation. This part is reflected in formula (4) and subsequent experimental verification.
(3) The paper is well organized, written in good English, however, a spell checking and minor improvements of style is recommended. Moreover, define abbreviations NCC and NSSD (p. 2) and ZNSSD (p. 3). Ref. 11 has no journal indicated. Writing the exponents in "in-line" form (such as 1e-3) is not appropriate.
Reply: All suggestions have been modified in the text. Thank you for your suggestions.
Reviewer 2 Report
The paper proposes a new color FA-GN algorithm, combining the characteristics of color images with three channels to measure complex deformation. Based on the linear change of brightness of three channels before and after deformation, this approach reduces the reference subset and improve the accuracy by avoiding second-order mapping. The paper is well packaged in its present form. Please address quantitative results of the experiments to highlight the significance in the abstract and in the conclusion.
Author Response
The paper proposes a new color FA-GN algorithm, combining the characteristics of color images with three channels to measure complex deformation. Based on the linear change of brightness of three channels before and after deformation, this approach reduces the reference subset and improve the accuracy by avoiding second-order mapping. The paper is well packaged in its present form. Please address quantitative results of the experiments to highlight the significance in the abstract and in the conclusion.
Reply: In the manuscript, numerical simulations have been conducted to compare and verify the advantages of our method over traditional methods in terms of computational accuracy. This can already demonstrate the effectiveness of the proposed method in complex deformation measurement. Subsequently, the compression deformation was measured using the method presented in this paper. The smoothness of the measured deformation field also demonstrates the correctness of the method proposed in this paper. The reviewer's suggestion is reasonable, but unfortunately, in real experiments, the difference between the proposed method and traditional methods cannot be quantified due to the unknown ideal deformation values.
Reviewer 3 Report
The manuscript is well written, well explained and substantiated. It is almost ready to be published. I only have small comments as follows:
1. Describe the acronyms NCC, NSSD, and ZNSSD.
2. Align the numbering of the equations to the left.
3. Align and number the equation that is between equations 7 and 8.
4. Correct the font type in Figure 5, Figure 6 Figure 7 and Figure 8.
5. Reference 28 comes after references 29 and 30; organize them well.
6. Increase the size of the numbering of the three-dimensional graphs (figures 7 and 8).
7. Explain Figure 11 in more detail.
8. Improve the layout of figure 11. Perhaps you could distribute the 4 images from part (o) in parts (a), (b), (c) and (d). Increase the size of the numbering and reposition the labels in blue.
Author Response
The manuscript is well written, well explained and substantiated. It is almost ready to be published. I only have small comments as follows:
- Describe the acronyms NCC, NSSD, and ZNSSD.
- Align the numbering of the equations to the left.
- Align and number the equation that is between equations 7 and 8.
- Correct the font type in Figure 5, Figure 6 Figure 7 and Figure 8.
- Reference 28 comes after references 29 and 30; organize them well.
- Increase the size of the numbering of the three-dimensional graphs (figures 7 and 8).
- Explain Figure 11 in more detail.
- Improve the layout of figure 11. Perhaps you could distribute the 4 images from part (o) in parts (a), (b), (c) and (d). Increase the size of the numbering and reposition the labels in blue.
Reply: Thank you for your suggestions. The formula numbers, font size in the figure, and numbers of references have been modified as much as possible. Regarding the content of the application section, as its purpose is to simply confirm the application of the method proposed in this paper in practical measurements and does not have practical significance, the data in Figure 11 was not analyzed in detail. The smoothness of the calculated results can already demonstrate the correctness of the measurement. In addition, the layout adjustment will take up more space, and we believe that the current layout will not affect readers' understanding.
Reviewer 4 Report
1. In the introduction of this manuscript, the mainstream sub-pixel matching algorithms are the forward accumulation Gauss-Newton method (FA-GN) [3] and the reverse combination Gauss-Newton method (IC-GN) [4]. Both have the same accuracy. In contrast, the former is more suitable for dealing with complex deformation or nonlinear light intensity changes, while the latter has higher computational efficiency. But, in my opinion, FA-GN and IC-GN have the same robustness for dealing with complex deformation or nonlinear light intensity changes.
2. There are some formatting problems in the reference. other grammar problems need to be checked again.
[11] Hijazi, A.; Al-Masri, A.; Rawashdeh, N., On the use of Bayer Sensor Color Cameras in Digital Image Correlation. 2022, 1-7.
3. In Fig.8, the maximum value and minimum value of the color bar should be double-checked.
Author Response
1.In the introduction of this manuscript, the mainstream sub-pixel matching algorithms are the forward accumulation Gauss-Newton method (FA-GN) [3] and the reverse combination Gauss-Newton method (IC-GN) [4]. Both have the same accuracy. In contrast, the former is more suitable for dealing with complex deformation or nonlinear light intensity changes, while the latter has higher computational efficiency. But, in my opinion, FA-GN and IC-GN have the same robustness for dealing with complex deformation or nonlinear light intensity changes.
Reply: Indeed, the description I made in the paper is not accurate. My original intention was that due to its basic structure, the FA-GN algorithm can choose different shape functions and light intensity change models. For example, if second-order shape function and nonlinear light intensity change model are selected in the FA-GN algorithm, the constructed matching algorithm can describe nonlinear deformation and nonlinear light intensity change to a certain extent. Due to its unique nature, the IC-GN algorithm usually uses ZNSSD function as the correlation function, while the shape function usually only describes linear deformation. In this way, IC-GN is generally applied in situations of linear light intensity changes and linear deformation.
In the manuscript, this sentence has been revised to: Compared to the IC-GN algorithm, FA-GN is more flexible in selecting shape function and correlation function.
2.There are some formatting problems in the reference. other grammar problems need to be checked again.
[11] Hijazi, A.; Al-Masri, A.; Rawashdeh, N., On the use of Bayer Sensor Color Cameras in Digital Image Correlation. 2022, 1-7.
Reply: I am very sorry for these low-level errors made during the submission process. The reference has been removed as it is not very important.
3.In Fig.8, the maximum value and minimum value of the color bar should be double-checked.
Reply: I have rechecked and run the written program. The values of the color bar in Figure 8 are correct. Since the program is written in MATLAB, its maximum and minimum values will be automatically adjusted based on the data in the 3D display. The peaks in Figure 8 (a) are 0.04 and 0.03, while the peaks in Figure 8 (b) are both 0.02. It precisely indicates that the method proposed has a smaller error compared to traditional matching method.
Round 2
Reviewer 4 Report
1. There is an error in Eq. 10.
2. For a similar task, an improved spatiotemporal correlation method for high-accuracy random speckle 3D reconstruction had been proposed [1]. It is recommended to provide the difference between the two works.
[1] Q Tang, C Liu, Z Cai, et al. An improved spatiotemporal correlation method for high-accuracy random speckle 3D reconstruction. Optics and Lasers in Engineering, 2018, 110: 54–62.
Round 3
Reviewer 4 Report
The author has addressed my comments. I recommend its publication in Sensors.